# TheoremQA: A Theorem-driven Question Answering Dataset

♠Wenhu Chen,*♥Ming Yin, ♠Max Ku, ♦Pan Lu, ♦Yixin Wan,
♠Xueguang Ma, ♥Jianyu Xu, ♥Xinyi Wang, ♦Tony Xia

University of Waterloo, Canada♠
University of California, Santa Barbara, United States♥
University of California, Los Angeles, United States♦

## Abstract

The recent LLMs like GPT-4 and PaLM-2 have made tremendous progress in solving fundamental math problems like GSM8K by achieving over 90% accuracy. However, their capabilities to solve more challenging math problems which require domain-specific knowledge (i.e. theorem) have yet to be investigated. In this paper, we introduce TheoremQA, the first theorem-driven question-answering dataset designed to evaluate AI models' capabilities to apply theorems to solve challenging science problems. TheoremQA is curated by domain experts containing 800 high-quality questions covering 350 theorems[1] from Math, Physics, EE&CS, and Finance. We evaluate a wide spectrum of 16 large language and code models with different prompting strategies like Chain-of-Thoughts and Program-of-Thoughts. We found that GPT-4's capabilities to solve these problems are unparalleled, achieving an accuracy of 51% with Program-of-Thoughts Prompting. All the existing open-sourced models are below 15%, barely surpassing the random-guess baseline. Given the diversity and broad coverage of TheoremQA, we believe it can be used as a better benchmark to evaluate LLMs' capabilities to solve challenging science problems.

## 1 Introduction

A long-standing goal of AI systems is to help human beings solve challenging problems, especially more domain-specific problems. To benchmark the progress towards this goal, researchers propose to evaluate AI systems' performance on different math word problem (WMP) datasets. In recent years, there has been a plethora of WMP datasets (Lu et al., 2023c), which we include in Table 1. Most of these datasets are meant for fundamental questions aimed at Grade 1-12 students on

---

* Authors ordered by contribution. Corresponding author email: wenhuchen@uwaterloo.ca

[1]e.g. Taylor's theorem, Lagrange's theorem, Huffman coding, Quantum Theorem, Elasticity Theorem, etc

a narrow subject. On the other hand, these datasets do not involve much domain-specific knowledge, aka **theorem**. Due to these two deficiencies, we believe that these datasets are not ideal to benchmark the existing powerful LLMs (Brown et al., 2020; Tamkin et al., 2022; Chen et al., 2021b; Chowdhery et al., 2022; Hoffmann et al., 2022; Taylor et al., 2022) due to their simplicity. In fact, on the popular GSM8K dataset (Cobbe et al., 2021), GPT-4 (OpenAI, 2023) and PaLM-2 (Google, 2023) both already achieved 92% accuracy. Similarly, we tested GPT-4 (OpenAI, 2023) on the subsets of several other listed datasets in Table 1 and observed 90+% accuracy in most cases. The only exception is MATH (Hendrycks et al., 2021) containing high-school math competition problems with SoTA performance around 50% (Zheng et al., 2023). However, MATH (Hendrycks et al., 2021) is focused on math skills rather than theorem.

In this paper, we propose the first theorem-driven QA dataset built on university-level theorems across Math, Physics, EE&CS, and Finance. The whole collection process takes two steps: (1) we first enumerate roughly 400 theorems in different subfields like algebra, number theory, graph theory, information theory, etc, (2) we ask domain experts to search for questions regarding these theorems from different sources like Internet and Textbooks. The domain experts will adjust these questions to ensure the answers follow the desired format for the ease of automatic evaluation. Through the careful construction process, we collected 800 high-quality question-theorem-answer triples as our final release version.

We evaluate a wide spectrum of instruction-finetuned language and code models including GPT (Brown et al., 2020), Claude (Bai et al., 2022), LLaMA (Touvron et al., 2023), Pythia (Biderman et al., 2023), CodeGen (Nijkamp et al., 2022), GLM (Zeng et al., 2022), StarCoder (Li et al., 2023), and CodeT5+ (Wang et al., 2023)

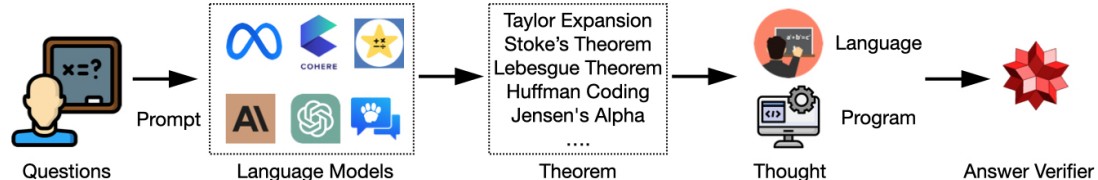

Figure 1: The overview of `TheoremQA` and the prompting strategies adopted.

| Dataset | Domain | Level | Source | Theorem |
|---|---|---|---|---|
| DRAW (Upadhyay and Chang, 2015) | Algebra | Elementary School | Generated | - |
| MAWPS (Koncel-Kedziorski et al., 2016) | Arithmetic | Elementary School | Generated | - |
| DRAW1K (Upadhyay and Chang, 2017) | Algebra | Elementary School | Generated | - |
| ASDiv (Miao et al., 2020) | Arithm/Algebra | Elementary School | Internet | - |
| SVAMP (Patel et al., 2021a) | Arithm/Algebra | Elementary School | ASDiv | - |
| Math23K (Wang et al., 2017) | Algebra | Elementary School | Internet | - |
| TabMWP (Lu et al., 2023b) | Arithm/Algebra | Elem./Middle School | Textbooks | NO |
| GSM8K (Cobbe et al., 2021) | Arithm/Algebra | Middle School | Annotated | NO |
| GEOS (Seo et al., 2015) | Geometry | Middle School | SAT | NO |
| Geometry3K (Lu et al., 2021a) | Geometry | Middle/High School | Textbooks | NO |
| GeoQA (Chen et al., 2021a) | Geometry | Middle/High School | Exam | NO |
| UniGeo (Chen et al., 2022a) | Geometry | Middle/High School | Textbooks | NO |
| ScienceQA (Lu et al., 2022) | Science | Middle/High School | Textbooks | NO |
| MATH (Hendrycks et al., 2021) | Math | High School | Competition | YES |
| AQuA (Ling et al., 2017) | Arithm/Algebra | University | GMAT/GRE | NO |
| MathQA (Amini et al., 2019) | Arithm/Algebra | University | AQuA | NO |
| MathQA-Python (Austin et al., 2021) | Arithm/Algebra | University | AQuA | NO |
| FinQA (Chen et al., 2021c) | Finance | University | CrowdSource | NO |
| TAT-QA (Zhu et al., 2021) | Finance | University | CrowdSource | NO |
| TheoremQA (Ours) | STEM | University | Internet+Expert | 350+ |

Table 1: List of existing Math and STEM QA datasets.

on our dataset. We adopt two prompting methods: Chain-of-Thoughts (CoT) (Wei et al., 2022b) and Program-of-Thoughs (PoT) (Chen et al., 2022b) to prompt the large language models. We also investigate how to infuse the theorem into the thought process of LLMs and how to present the multimodal inputs to the LLMs.

In the course of our experiments, several notable observations were made. First, GPT-4 (OpenAI, 2023) significantly outperformed all existing models, reaching an accuracy level of 51% when combined with Program-of-Thoughts prompting. Trailing behind GPT-4, the second most effective model was ChatGPT, achieving an accuracy of 35% through the same prompting method. Additionally, our human evaluation determined that half of GPT-4's errors are caused by minor mistakes like calculation errors, rounding errors, etc. We believe these errors could be easily rectified with a more deliberate prompting strategy or human intervention. This suggests that there is still significant headroom for GPT-4 to achieve with more deliberate prompting strategies. Secondly, we found that all open-source, instruction-tuned language and code models scored below 15% in accuracy, barely exceeding the random guess baseline of 10%. Our human evaluation reveals that open-source models like Alpaca are making errors mainly due to their ignorance of the theorem, where 90% of the errors are not rectifiable. This stark gap between GPT and open-source models suggests that further enhancement strategies, such as science-focused pre-training or fine-tuning, should be considered to narrow the performance disparity. Thirdly, we explored the potential to do theorem-augmented generation. However, the simple strategy of concatenation did not yield a significant improvement. We conjecture that a more complex integration strategy may be needed to achieve more gains. Lastly, we examined the performance of various multi-modal instruction-tuned models on the multimodal subset of the `TheoremQA` dataset. Surprisingly, these models did not demonstrate significant performance gains over their text-only counterparts. This is mainly due to the unnaturalness of the image, which consists of lots of diagrams and text. Such images are not well

captured by existing visual encoder models.

To sum up, our contributions are three folds:

- We propose the first theorem-driven question-answering dataset to understand LLMs' capabilities to apply science theorems.

- We comprehensively evaluate a wide spectrum of 16 LLMs on TheoremQA.

- We perform different analyses in the theorem integration and multimodal understanding aspects to provide detailed insights.

## 2 Related Work

### 2.1 Math Word Problems

Mathematical reasoning skills are crucial for general-purpose intelligent systems, garnering significant interest from the research community. In the past, studies have explored the ability of NLP models to solve arithmetic and algebraic problems (Hosseini et al., 2014; Koncel-Kedziorski et al., 2015; Roy and Roth, 2015; Ling et al., 2017). More recently, researchers have introduced increasingly challenging datasets (Saxton et al., 2019; Miao et al., 2020; Amini et al., 2019; Hendrycks et al., 2021; Lu et al., 2021b; Patel et al., 2021b) aimed at enhancing difficulty, diversity, and adversarial robustness. LiLA (Mishra et al., 2022) proposes to assemble a vast collection of mathematical datasets into a single, unified dataset. LiLA also annotates Python programs as target outputs for solving mathematical problems. However, the existing datasets were mostly focused on grade school simple mathematics. To further investigate the LLMs' capabilities to assist humans to solve challenging math problems, we propose TheoremQA as the first benchmark to enable research in this direction.

### 2.2 Large Language Models

In recent years, there has been a surge of research and development in the area of large language models (LLMs) that have significantly advanced the field of natural language processing. GPT-3 (Brown et al., 2020) demonstrated a strong capability to perform few-shot predictions, where the model is given a description of the task in natural language with few examples. By using human-feedback reinforcement learning, Instruct-GPT (Ouyang et al., 2022) has shown its unprecedented capabilities to follow human instructions.

Scaling model size, data, and computing are crucial to enable this learning ability. Later, Rae et al. (2021); Chowdhery et al. (2022); Zhang et al. (2022); Touvron et al. (2023); Chen et al. (2021b) have proposed to train different types of LLMs with different training recipes. The capability to follow few-shot exemplars to solve unseen tasks is not existent on smaller LMs, but only emerges as the model scales up (Wei et al., 2022a). More recently, GPT-4 (OpenAI, 2023) shows tremendous progress on lots of complex reasoning tasks spanning mathematics, coding, vision, medicine, law, psychology, and more. Bubeck et al. (2023) shows that GPT-4 is already demonstrating more general intelligence than previous AI models. To further validate GPT-4's capability to solve challenging reasoning tasks, we propose TheoremQA as the new benchmark to further understand LLMs' upper limit.

### 2.3 Reasoning with Large Language Model

To better unleash large language models' capabilities to solve complex reasoning tasks. Chain-of-Thought Prompting (Wei et al., 2022b; Kojima et al., 2022; Wang et al., 2022) was proposed, which aims at prompting the large language models to generate the 'thought process' before outputting the answer. Later on, several other works (Drozdov et al., 2022; Zhou et al., 2022; Nye et al., 2021) also propose different approaches to utilize LLMs to solve reasoning tasks by allowing intermediate steps. Our method can be seen as an extension to CoT by leveraging an extra step of symbolic execution. Another line of work (Gao et al., 2022; Chen et al., 2022b) was proposed to adopt Python programs as the demonstration for the 'thought process' to solve different reasoning tasks.

## 3 Dataset

Our dataset collection pipeline contains two steps:

**Theorem Enumeration** Our aim was to encompass a wide range of theorems. To this end, we began by prompting Large Language Models (LLMs), specifically GPT-4 (OpenAI, 2023), to enumerate popular subfields in Mathematics, Physics, Finance, and Electrical Engineering & Computer Science. The covered subfields are listed in Figure 4. Subsequently, we prompted GPT-4 to propose plausible university-level theorems relevant to these subfields. For instance, within the 'Calculus' subfield, GPT-4 might suggest the 'Intermediate Value Theorem', 'Rolle's Theorem', and so on. After gath-

ering an extensive list of theorems, we assembled a team of domain experts (holders of Masters and PhDs in Statistics, Electrical Engineering, Computer Science, and Finance) to refine the theorem inventory and supplement any omitted theorems. Ultimately, we collected approximately 400 theorems, encapsulating a diverse range of topics within these fields. We then delegated these theorems to nine domain experts, instructing them to locate question/answer pairs from varied sources. During the annotation process, a small number of theorems were discarded due to their evaluation complexity.

**Question Annotation** Our problems were sourced from websites, books, or devised by the experts themselves. One challenge we encountered was the potential for questions found online to have been included in the training data. To mitigate this 'data contamination' issue, we encouraged domain experts to modify these questions. Another challenge arose from questions with answers in symbolic form, matrix form, figure form, etc. These presented significant obstacles for our automatic evaluation. To overcome this, we instructed domain experts to alter the question so the answer would be limited to the following forms: (1) integer, (2) float, (3) list of integers/floats, (4) boolean, and (5) multiple-choice options. For instance, if the original question concerned a matrix, we would revise it to ask about the trace of the answer matrix. This modification significantly streamlined the evaluation process. An example of this can be found in Figure 2.

**Dataset Statistics** Finally, we collected a total of 800 questions over 354 theorems. Specifically, there are 199 Math theorems, 52 Physics theorems, 55 Finance theorems, and 48 CS&EE theorems. There are 442 Math questions, 146 CS&EE questions, 131 physics questions, and 81 Finance questions. We show the answer-type distribution in Figure 3. To further enhance the multimodality aspect of TheoremQA, we also include 51 questions with image input (diagrams), where the model needs to understand the visual input to answer questions.

The majority of the questions in TheoremQA have float and integer as the answers, which is more realistic than the existing multi-choice datasets like ScienceQA (Lu et al., 2022) or AQuA QA (Ling et al., 2017). Therefore, the models are unlikely to take shortcuts to achieve high accuracy.

**Question**: Please use the Stoke's therorem to evaluate $\iint_S curl\vec{F} \cdot d\vec{r}$ where $\vec{F} = z^2\vec{i} - 3xy\vec{j} + x^3y^3\vec{k}$ and $S$ is the part of $z = 5 - x^2 - y^2$ above the plane $z=1$. S is oriented upwards.

**Stoke's Theorem**: Let S be an oriented smooth surface that is bounded by a simple, closed, smooth boundary curve C with positive orientation. Also let $\vec{F}$ be a vector. We can compute the integral as follows:

$$\int_C \vec{F} \cdot d\vec{r} = \iint_S curl\vec{F} \cdot d\vec{S}$$

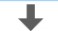

Answer: 0, Type: Float

**Quesiton**: Let W(t) be the standard Brownian motion. Find the probability of $P(W(1) + W(2) > 2)$.

**Winer's Process** The Wiener process $W_t$ is characterised by the following properties: $W$ has independent increment. For every $t > 0$, the future increment $W_{t+u} - W_t$ are independent from the past $W_t$. W has Gaussian increments, $W_{t+u} - W_t$ has Gaussian distribution $\mathcal{N}(0, u)$.

Answer: 0.186, Type: Float

Figure 2: Examples from TheoremQA. The first question requires the usage of Stoke's theorem to transform the double integral into a line integral. The second question requires knowing the properties of Wiener's process.

**Human-Level Performance** To provide a rough but informative estimate of human-level performance. We randomly select 20 questions and assign these questions to the 4 Math&CS undergraduate students (average GPA) who have taken the required courses regarding these questions. The participants are given 24 hours with internet access to solve these questions. The four undergraduate students achieve 12/20, 15/20, 18/20, and 19/20 scores on these randomly sampled questions. From this experiment, we are more confident that an expert-level performance should be 100%.

## 4 Method

Our method for addressing these demanding questions in the TheoremQA is comprised of several distinct modules, as outlined in Figure 1:

**Prompting** We utilize two established prompting strategies:

- Chain-of-Thought Prompting (Wei et al.,

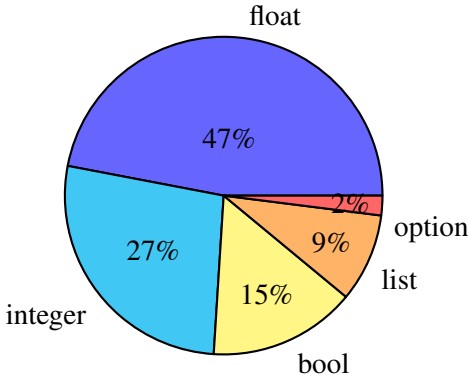

Figure 3: Answer type distribution in `TheoremQA`.

2022b): This strategy prompts the language model to initially generate a step-by-step thought process, eventually leading to the final answer.

- Program-of-Thought Prompting (Chen et al., 2022b; Gao et al., 2022): This strategy prompts the language model to progressively generate a program. The final answer is then derived by executing this program.

By delegating computational tasks to an external executor, the problem-solving process is considerably enhanced in its reliability. This improvement results in remarkable advancements in existing math datasets being reported in (Chen et al., 2022b).

**Answer Extraction** We observed that parsing the output from Large Language Models (LLMs) can be challenging due to two main issues: (1) The answer is often embedded within a sentence, making it difficult to extract using regular expressions, and (2) The answer may not be normalized, such as 'pi / 3' or '2*10 - e', which complicates comparison with the ground truth. To tackle these problems, we initially employ ChatGPT to identify the answer span within the model's output, then forward this string to WolframAlpha (Inc.) for normalization into a float, integer, or list.

**Theorem Augmentation** We explored the potential of enhancing large language models with retrieved theorem descriptions to assess their effect on performance. One approach is to retrieve descriptions of the given theorems from the Internet to supplement the LLMs' output. Another experiment involved prompting GPT-4 to generate text descriptions of the theorem, which are then used as an additional augmentation signal.

**Multimodal Input** A small portion of our data (50 instances) includes images, such as diagrams, as supplemental input, particularly in geometry questions. Since current LLMs don't support such multimodal inputs, we propose a solution: to employ captions like Chameleon (Lu et al., 2023a). These captions describe the image and are then appended to the LLMs' output as an additional signal.

## 5 Experiments

### 5.1 Model Descriptions

In our experiments, we mainly investigate the following models:

- GPT3/3.5/ChatGPT/GPT4: These are instruction-tuned models from OpenAI[2].

- PaLM-2: This is the instruction-tuned model from Google (Google, 2023).

- Claude-v1/Claude-instant: These are instruction-tuned models from AnthropicAI[3].

- Alpaca-13B: This model is based on the LLaMA (Touvron et al., 2023). Alapaca is instruction-tuned by the 52K data generated from GPT-4.

- Vicuna-13B: This model is based on the LLaMA (Touvron et al., 2023). Vicuna is instruction-tuned by the 100K ShareGPT data generated by different GPT-based models.

- OpenAssistant-12B: This model is based on Pythia (Biderman et al., 2023). The model is instruction-tuned by OpenAssistant data[4].

- MOSS-instruct-16B: This model is based on CodeGen (Nijkamp et al., 2022), which is further instruction-tuned with instruction following dataset distilled from GPT.[5]

- StarChat-16B: This model is based on StarCoder (Li et al., 2023). StartChat is being instruction-tuned on OpenAssistant data[6] and ShareGPT data.

[2]https://openai.com/
[3]https://www.anthropic.com/index/introducing-claude
[4]https://open-assistant.io/
[5]https://txsun1997.github.io/blogs/moss.html
[6]https://open-assistant.io/

| Mathematics | | Physics | | Finance | |
|---|---|---|---|---|---|
| Calculus | 103 | Kinetics | 30 | Economics | 22 |
| Combinatorics | 57 | Electromagnetism | 21 | Quantitive methods | 14 |
| Algebra | 52 | Atomic physics | 11 | Derivatives | 14 |
| Mathematical analysis | 42 | Wave | 8 | Fixed income | 11 |
| Number theory | 29 | Optics | 8 | management | 10 |
| Geometry | 27 | Condensed matter | 8 | Investments | 10 |
| Numerical analysis | 24 | Particle | 6 | Total | 81 |
| Statistics | 24 | Statistical physics | 6 | | |
| Complex analysis | 21 | Relativity | 7 | **CS & EE** | |
| Probability theory | 19 | Celestial mechanics | 6 | Signal processing | 47 |
| Stochastic process | 16 | Thermodynamics | 5 | Graph theory | 34 |
| Group theory | 11 | Quantum | 5 | Information theory | 29 |
| Functional analysis | 10 | Classic mechanics | 5 | Computer networking | 23 |
| Real analysis | 7 | Fluid mechanics | 5 | Machine learning | 13 |
| Total | 442 | Total | 131 | Total | 146 |

Figure 4: Subfields of `TheoremQA` under Math, Physics, Engineering, and Finance.

- InstructCodeT5+: This model is based on CodeT5+ (Wang et al., 2023). Instruct-CodeT5+ is further instruction-tuned on Code Alpaca data[7] to follow instructions.

## 5.2 Main Results

We demonstrate our main results on Table 2. We will summarize different findings in the following:

**Closed-source Models** For GPT-3 (text-davinci-002) and GPT-3.5 model, since these two models are not Chat-based models, we need to demonstrate one example ensure to help them generate outputs of the desired format. With CoT prompting, GPT-3 (text-davinci-002) and GPT-3.5 models are only achieving 16.6% and 22.8% accuracy. By adopting the program as the intermediate reasoning form, both models can gain reasonable improvements. For Claude-v1, we found that it is matching the performance of GPT-3.5. ChatGPT outperforms GPT-3.5 and Claude-v1 significantly by 8%, which indicates ChatGPT's capabilities to perform complex numerical reasoning. GPT-4 is the strongest model being evaluated, which beats all the rest models by a huge margin. With Chain-of-Thoughts prompting, GPT-4 can outperform ChatGPT by 13%. With Program-of-Thoughts prompting, GPT-4 can outperform ChatGPT by 16%. Though some other models have shown to match GPT-4 on simple tasks, GPT-4's capability to solve challenging tasks seems unparalleled.

[7]https://github.com/sahil280114/codealpaca

**Open-source Models** For the open-source models, we found that their performance is much behind. To better understand their accuracy, we also provide the random-guess baseline of 10%. We test both prompting strategies, however, their results consistently lie in the range of 10-14%. The results indicate that these open-source LMs are still struggling with more complex mathematical reasoning tasks in `TheoremQA`. Given that ChatGPT of a similar size is able to achieve much higher performance, we believe the parameter size is not the only cause. There is still a significant amount of effort during pre-training or supervised fine-tuning to instill enough scientific knowledge into the models' parameters to close the gap.

**Program of Thoughts Analysis** From Table 2, we observe that PoT brings consistent improvement over CoT on GPT-* models. Different GPT-* models can normally yield a gain of 5-8% accuracy. In contrast, Claude-v1 and StarChat are almost obtaining the same accuracy. To better analyze where the gains are coming from, we plot Figure 5 to understand how many of generated Python programs are actually 'executable'. As can be seen, both StarChat and CodeT5+ are having trouble generating 'runnable' programs with only 40% programs being executable. Claude-v1 is able to increase the validity of the generated programs to 60%. In contrast, GPT3.5 and ChatGPT can further increase the ratio to around 80%. GPT-4 is extremely accurate in generating programs, where 92% of the generated programs are runnable. Such a high executable ratio explains why the gain brought to GPT-* model

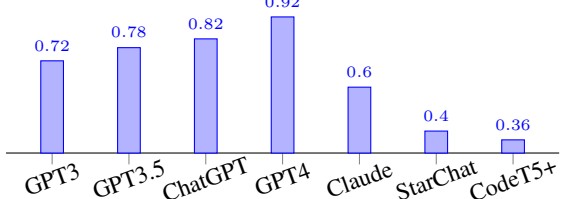

Figure 5: Ratio of Executable Python Program of different models with PoT prompting.

is much higher than Claude-v1 and StarChat.

### 5.3 Additional Result

**Theorem Augmentation** We also investigate whether feeding theorem as an additional text condition would help the model better solve the problem. Specifically, we ask GPT-4 to generate a paragraph to describe the theorem, which we post-processed to ensure correctness. We feed the theorem in the prompt to different language models to see the performance change and plot our findings in Table 3. For all the evaluated scenarios, we found that the improvement is limited to within 1%. Unlike the Text or KB knowledge, theorem knowledge is more abstract and symbolic, simply concatenating the theorem definition is not enough. We believe a more sophisticated augmentation scheme is needed to truly help the model understand and apply the theorems to solve problems.

**Multimodal Questions** Our aim was to assess how effectively the current method could tackle multimodal questions (those with image inputs) in the `TheoremQA` dataset. An example is illustrated in Figure 6, where an image is converted into 'captions' by BLIP (Li et al., 2022). We graphed the results from over 50 multimodal question subsets in Figure 7. Notably, this subset posed substantial challenges; none of the models were able to achieve an accuracy rate of 10%. This is primarily due to information loss during the captioning process.

In light of this, we conducted further evaluations on two multimodal instruction-tuned models, LLaVA-13B (Liu et al., 2023) and VisualGLM-6B (Zeng et al., 2022)[8]. These models utilize a visual encoder (either CLIP (Radford et al., 2021) or BLIP (Li et al., 2022)) to encode image input, which is then integrated with language models for multimodal conversation. However, these models demonstrated performance similar to their text-only

[8]https://github.com/THUDM/VisualGLM-6B

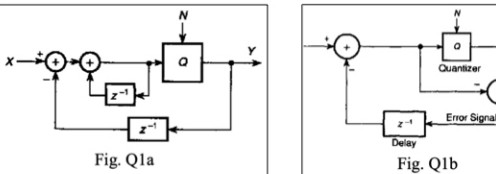

Question: Are the circuits shown in Fig. Q1a and Fig. Q1b are identical in terms of the Tranfer functions.

BLIP Caption: a diagram of a block diagram with a block diagram and a block diagram.

Figure 6: An example of Multimodal question.

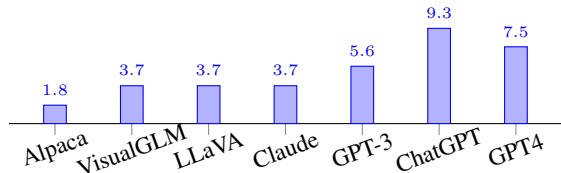

Figure 7: Accuracy on the Multimodal Question Subset

equivalent, Alpaca, with the addition of a visual encoder not significantly enhancing the results. We hypothesize that the current visual encoding modules may not be suited for representing these diagrammatic images, resulting in these less than ideal outcomes. We believe these multimodal questions remain a challenge for the research community, and we eagerly anticipate further advancements in addressing these multimodal scientific questions.

**Error Analysis** We conduct detailed error analysis on 200 erroneous cases from different models to analyze their error distribution. Specifically, we pick GPT4, ChatGPT and Alpaca to understand their error sources. We include the following error types: (E1) the model does not even know this theorem, (E2) the model does know the theorem, but uses the wrong formula or algorithm, (E3) the model knows the theorem and the formula, the error is only caused by minor calculation mistakes. The severity of the error decrease as the error number increases. We plot our findings in Figure 8, where the bar indicate the percentage of specific error types. We can observe that almost half of the errors made by GPT4 are non-critical with caused by minor calculation mistakes. This error analysis suggests that there is a still a significant headroom for GPT4 to improve with more deliberate prompting strategies or human intervention to mitigate these minor errors. In contrast, Alpaca's errors are mainly caused by not knowing the theorem at all.

| Model | Integer | Float | Option | List | Bool | Math | CS&EE | Physics | Finance | All |
|---|---|---|---|---|---|---|---|---|---|---|
| Random Guess | 0 | 0 | 38.9 | 0 | 65.5 | 10.0 | 24.7 | 0 | 4.9 | 10.5 |
| Chain of Thoughts (CoT) | | | | | | | | | | |
| GPT-3 | 11.6 | 11.7 | 27.8 | 6.8 | 46.6 | 15.8 | 34.2 | 2.3 | 12.3 | 16.6 |
| GPT-3.5 | 13.0 | 14.3 | 50.0 | 13.7 | 69.8 | 22.6 | 36.3 | 7.6 | 23.5 | 22.8 |
| ChatGPT | 32.4 | 22.3 | 50.0 | 20.5 | 55.2 | 31.0 | 41.1 | 16.8 | 28.4 | 30.2 |
| GPT-4 | 40.3 | 36.7 | 66.7 | 37.0 | 74.6 | 43.9 | 50.6 | 30.5 | 51.4 | **43.8** |
| PaLM-2 | 26.4 | 22.8 | 61.1 | 23.3 | 71.6 | 31.0 | 47.3 | 19.8 | 27.2 | 31.8 |
| Claude-v1 | 18.1 | 19.4 | 27.8 | 15.1 | 61.2 | 21.7 | 42.5 | 13.7 | 28.4 | 24.9 |
| Cluade-instant | 19.9 | 16.7 | 44.4 | 17.8 | 53.4 | 21.5 | 36.3 | 14.5 | 27.2 | 23.6 |
| Alpaca (13B) | 11.1 | 6.9 | 27.8 | 2.7 | 45.7 | 12.9 | 27.4 | 3.8 | 9.9 | 13.5 |
| Vicuna (13B) | 8.8 | 6.9 | 16.7 | 2.7 | 45.7 | 12.2 | 24.0 | 3.1 | 12.3 | 12.9 |
| OpenAssistant (12B) | 8.3 | 5.0 | 22.2 | 1.4 | 37.9 | 10.2 | 25.0 | 0 | 4.9 | 10.7 |
| MOSS (16B) | 8.8 | 5.4 | 24.2 | 2.4 | 44.2 | 11.3 | 28.4 | 1.6 | 8.9 | 12.2 |
| StarChat (16B) | 7.9 | 4.9 | 22.3 | 1.9 | 44.1 | 10.7 | 23.5 | 0.6 | 6.8 | 11.6 |
| Program of Thoughts (PoT) | | | | | | | | | | |
| GPT-3 | 17.1 | 15.9 | 22.2 | 9.6 | 49.1 | 23.3 | 25.4 | 8.4 | 17.3 | 20.6 |
| GPT-3.5 | 23.6 | 19.9 | 50.0 | 21.9 | 61.2 | 26.7 | 41.1 | 14.5 | 30.9 | 27.8 |
| ChatGPT | 31.0 | 35.0 | 38.9 | 21.9 | 54.3 | 35.7 | 35.6 | 26.7 | 49.4 | 35.6 |
| GPT-4 | 44.4 | 50.7 | 66.7 | 39.7 | 78.4 | 52.0 | 51.4 | 45.8 | 66.7 | **52.4** |
| Claude-v1 | 17.1 | 21.8 | 33.3 | 6.9 | 62.5 | 23.1 | 37.5 | 17.1 | 28.4 | 25.9 |
| StarChat (16B) | 7.7 | 6.1 | 0.0 | 3.0 | 43.5 | 13.6 | 17.6 | 5.1 | 5.1 | 11.3 |
| InstructCodeT5+ (16B) | 8.9 | 6.3 | 0.0 | 6.9 | 45.2 | 13.8 | 17.9 | 4.2 | 5.1 | 11.6 |

Table 2: Results for CoT and PoT prompting on `TheoremQA`. We report the accuracy over different fine-grained question types and scientific fields.

| Model | Method | Theorem | All |
|---|---|---|---|
| ChatGPT | CoT | - | 30.2 |
| ChatGPT | CoT | + | 30.8 |
| Claude-v1 | CoT | - | 24.9 |
| Claude-v1 | CoT | + | 25.4 |
| ChatGPT | PoT | - | 35.6 |
| ChatGPT | PoT | + | 35.8 |
| Alpaca-13B | CoT | - | 13.5 |
| Alpaca-13B | CoT | + | 14.2 |

Table 3: Results for CoT and PoT prompting with additional theorem conditions.

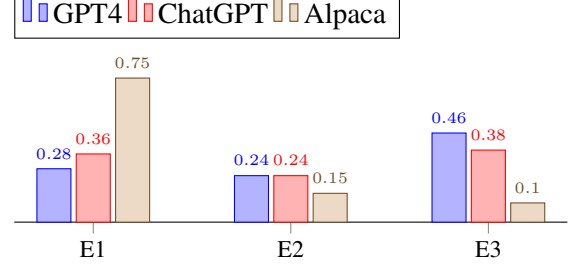

Figure 8: Error Analysis for GPT4, ChatGPT and Alpaca. Error severity: E1 > E2 > E3

**Case Study**   We list a few successful and failed examples generated by GPT-4 in Figure 9 to do a side-by-side comparison between chain-of-thoughts prompting and program-of-thoughts prompting. In the first example, the question is regarding 'orthogonal projection theorem'. As can be seen, Chain-of-Thoughts prompting requires a very long paragraph to generate the results. We prompted GPT-4 a few times with the same input and the results seem unstable. Sometimes the model will make tiny computation mistakes in the middle to derive the wrong answer. In contrast, the program solution is brief and concise, which leads to rather stable outputs. For the second ex-

ample, the computation requires 'for loop' to iteratively compute delta values for Riemann Sum. We found that such problems are also more natural for programs to solve. Through these examples, we can see GPT-4's unprecedented capabilities to solve these difficult math problems even without any demonstration or hints.

We also show some examples in Figure 9 to compare the results of CoT and PoT prompting. We can see that the PoT can significantly shorten the output sequence length. By leveraging the additional tool, PoT is able to significantly lower the task difficulty.

**Question:** Find the orthogonal projection of [9, 0, 0, 0] onto the subspace of R^4 spanned by [2, 2, 1, 0] and [-2, 2, 0, 1].

The orthogonal projection of a vector onto a subspace W spanned by a set of vectors {v1, v2, ..., vn} is given by the sum of the projections onto each of these vectors.
In your case, the vector is v = 9e_1 = [9, 0, 0, 0] and the subspace W is spanned by u = [2, 2, 1, 0] and w = [-2, 2, 0, 1].
The projection of v onto u is given by:
proj_u(v) = ((v•u) / ‖u‖^2) * u
where "•" denotes the dot product, and ‖u‖ is the norm of u.
And the projection of v onto w is given by:
proj_w(v) = ((v•w) / ‖w‖^2) * w
The total projection onto the subspace W is then given by the sum of these individual projections:
proj_W(v) = proj_u(v) + proj_w(v)
Now let's compute these values:
The dot product v•u = (9 * 2) + (0 * 2) + (0 * 1) + (0 * 0) = 18
The norm ‖u‖^2 = (2^2 + 2^2 + 1^2 + 0^2) = 9
So, proj_u(v) = ((18) / 9) * [2, 2, 1, 0] = [4, 4, 2, 0]
Similarly, the dot product v•w = (9 * -2) + (0 * 2) + (0 * 0) + (0 * 1) = -18
The norm ‖w‖^2 = (-2^2 + 2^2 + 0^2 + 1^2) = 9
So, proj_w(v) = ((-18) / 9) * [-2, 2, 0, 1] = [4, -4, 0, -2]
The total projection onto the subspace W is then given by the sum of these individual projections:
proj_W(v) = proj_u(v) + proj_w(v) = [4, 4, 2, 0] + [4, -4, 0, -2] = [8, 0, 2, -2]
So, the orthogonal projection of 9e_1 onto the subspace of R^4 spanned by [2, 2, 1, 0] and [-2, 2, 0, 1] is [8, 0, 2, -2].

Chain of Thoughts ✅

```python
def solve():
    import numpy as np

    # Define the given vectors
    e1 = np.array([1, 0, 0, 0])
    v1 = np.array([2, 2, 1, 0])
    v2 = np.array([-2, 2, 0, 1])

    # Calculate the orthogonal projection
    A = np.column_stack((v1, v2))
    proj_matrix = A @ np.linalg.inv(A.T @ A) @ A.T
    proj_e1 = proj_matrix @ (9 * e1)

    return proj_e1.tolist()

print(solve())
```

Program of Thoughts ✅

**Question:** Let rectangle R = [1, 2.5] * [1, 2]. Calculate the Riemann Sum S_{3,2} for $\int \int_{R}$ xy dA for the integral, using the lower-left vertex of rectangles as sample points.

The given region R is a rectangle from [1, 2.5] in x and [1, 2] in y. We are asked to find the Riemann sum S_{3,2} which means we will partition the interval in x into 3 equal subintervals and the interval in y into 2 equal subintervals.

The width of each subinterval in x, Δx, is (2.5 - 1) / 3 = 0.5. The height of each subinterval in y, Δy, is (2 - 1) / 2 = 0.5.
Now, we are sampling from the lower left corner of each subrectangle. This means we'll be evaluating our function f(x, y) = xy at the points (x, y) where x ranges over {1, 1.5, 2}, and y ranges over {1, 1.5}.

The Riemann sum is given by:
S_{3,2} = Δx * Δy * Σ f(x_i, y_j)
where the sum Σ is over all pairs (x_i, y_j) from the lower left vertices of the subrectangles.
= 0.5 * 0.5 * [(1·1 + 1·1.5 + 1.5·1 + 1.5·1.5) + (1.5·1 + 1.5·1.5 + 2·1 + 2·1.5)]
= 0.25 * [3.5 + 5]
= 0.25 * 8.5 = 2.125

So, the Riemann sum S_{3,2} for the double integral over R of f(x, y) = xy with respect to area, using the lower-left vertex of each subrectangle as the sample point, is 2.125.

Chain of Thoughts ❌

```python
def solve():
    # Define the limits of integration
    x_min, x_max = 1, 2.5
    y_min, y_max = 1, 2
    # Define the number of rectangles in each direction
    num_rectangles_x = 3
    num_rectangles_y = 2
    # Calculate the width and height of each rectangle
    delta_x = (x_max - x_min) / num_rectangles_x
    delta_y = (y_max - y_min) / num_rectangles_y
    # Initialize the Riemann Sum
    riemann_sum = 0
    # Iterate through the rectangles
    for i in range(num_rectangles_x):
        for j in range(num_rectangles_y):
            # Calculate the lower-left vertex of the rectangle
            x = x_min + i * delta_x
            y = y_min + j * delta_y
            # Calculate the value of the function at
            f_xy = x * y
            # Add the contribution to the Riemann Sum
            riemann_sum += f_xy * delta_x * delta_y
    return riemann_sum
```

Program of Thoughts ✅

Figure 9: Case Study of GPT-4 generation with both prompting strategies.

## 6 Conclusion

In this paper, we propose the first theorem-driven science question-answering dataset and evaluate different LLMs on it. Though GPT-4 can achieve strong performance on our new dataset, the existing open-source LLMs are still struggling to achieve reasonable performance. We conjecture it is essential to leverage more science-related pre-training or fine-tuning to close the gap. On the hand, we found that the multimodal science questions are still extremely challenging for the existing visual LLMs. We believe more specialized visual encoding models are needed to better represent diagrams in these science questions.

## Limitations

In this work, we explore the possibilities to utilize different large language models to solve challenging theorem-driven questions. There are still some limitations: (1) our answer extraction is still not perfect. There are some cases where our answer extractor is not able to locate the answer. Thus the final accuracy is still an approximate lower bound. (2) in our dataset collection, we specifically avoid the hard-to-evaluate cases where the answer is a formula, figure, or a matrix. Our choice of the questions can be biased in terms of evaluating the overall ability. (3) in the multimodal questions in TheoremQA, we have investigated different existing models but none of them succeed in achieving reasonable performance.

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
