# OpenReview forum: "TheoremQA: A Theorem-driven Question Answering Dataset"
_EMNLP/2023/Conference — EMNLP 2023 Main_

### Official Review · Reviewer_aN4a · 2023-08-03

**Soundness:** 4

**Excitement:**

4: Strong: This paper deepens the understanding of some phenomenon or lowers the barriers to an existing research direction.

**Paper Topic And Main Contributions:**

The paper introduces TheoremQA, a theorem-driven question-answering dataset designed to evaluate AI models' capabilities to apply theorems to solve challenging science problems. The dataset covers a wide range of theorems from various fields such as Math, Physics, Electrical Engineering & Computer Science, and Finance.
Main Contributions:
1. The authors propose a theorem-driven question-answering dataset to understand Large Language Models' (LLMs) capabilities to apply science theorems.
2. They comprehensively evaluate a wide spectrum of 16 LLMs on TheoremQA.
3. They perform different analyses in the theorem integration and multimodal understanding aspects to provide detailed insights.

**Reasons To Accept:**

1.The proposed dataset, TheoremQA, is the first of its kind to focus on theorem-driven question-answering. It covers a wide range of theorems from various fields such as Math, Physics, Electrical Engineering & Computer Science, and Finance. This dataset can serve as a valuable resource for researchers in the field.
2. The authors have conducted a thorough evaluation of a wide spectrum of 16 LLMs on TheoremQA. Their comprehensive analysis provides valuable insights into the performance of these models and their ability to integrate theorems and understand multimodal inputs.
3. The methodology adopted by the authors is sound and well-justified. They have used innovative prompting strategies, including Chain-of-Thoughts (CoT) and Program-of-Thoughts (PoT), to guide the LLMs.

**Reasons To Reject:**

The paper does not provide detailed examples of the prompts used in the experiments. Including such examples, particularly in the appendix, would have been beneficial for readers to better evaluate the appropriateness and effectiveness of the prompting strategies employed by the authors. Without these examples, it is challenging to fully assess the validity of the authors' claims.

**Reproducibility:**

5: Could easily reproduce the results.

**Reviewer Confidence:**

3: Pretty sure, but there's a chance I missed something. Although I have a good feel for this area in general, I did not carefully check the paper's details, e.g., the math, experimental design, or novelty.

---

> ### Author Rebuttal · Authors · 2023-08-25
>
> Thanks a lot for the recognition of our paper.
>
> - The paper does not provide detailed examples of the prompts used in the experiments. Including such examples, particularly in the appendix, would have been beneficial for readers to better evaluate the appropriateness and effectiveness of the prompting strategies employed by the authors. Without these examples, it is challenging to fully assess the validity of the authors' claims.
>
> Thanks for the reminder. We will update our paper to include the used prompts and exemplars in our revision. The prompts and exemplars will also be released in our Github repo to help other researchers replicate our results.

---

### Official Review · Reviewer_CivJ · 2023-08-03

**Typos Grammar Style And Presentation Improvements:** None
**Soundness:** 4

**Excitement:**

4: Strong: This paper deepens the understanding of some phenomenon or lowers the barriers to an existing research direction.

**Missing References:**

None

**Paper Topic And Main Contributions:**

This paper introduces TheoremQA, a new question answering dataset for evaluating AI models' ability to apply theorems to solve challenging science problems. The main contributions are:

1. Proposes the first theorem-driven QA dataset spanning math, physics, engineering, and finance with 800 questions covering over 350 theorems.
2. Comprehensively evaluates 16 large language models with different prompting strategies.
3. Analyzes errors and suggests room for improvement.

Overall, the paper introduces a valuable new benchmark for testing knowledge application in scientific reasoning, while providing insights into current model capabilities and limitations. The dataset and analyses should spur further research.

**Questions For The Authors:**

1. An open question is: what are the possible reasons for the poor performance of current open-source models? Furthermore, what are the directions that have already been identified for potential improvements?

**Reasons To Accept:**

Motivation:

- Addresses an important open problem - evaluating scientific reasoning capabilities of AI systems, which requires applying knowledge.
- Proposes a new benchmark dataset to advance progress in this direction.

Experiments:

- Thorough comparative evaluation of 16 state-of-the-art models with different prompting strategies.
- Rigorous analysis of results reveals capabilities and limitations of current models.

Writing:

- Clearly presented. Does a good job explaining the dataset, experiments, results and analyses.
- Provides useful insights that can guide future research to overcome limitations.
- Shares data to enable further analysis.

**Reasons To Reject:**

This is a resource paper, and it excels in all aspects. I believe it deserves acceptance.

**Reproducibility:**

4: Could mostly reproduce the results, but there may be some variation because of sample variance or minor variations in their interpretation of the protocol or method.

**Reviewer Confidence:**

4: Quite sure. I tried to check the important points carefully. It's unlikely, though conceivable, that I missed something that should affect my ratings.

---

> ### Author Rebuttal · Authors · 2023-08-25
>
> Thanks a lot for the recognition of our paper.
>
> - "An open question is: what are the possible reasons for the poor performance of current open-source models? Furthermore, what are the directions that have already been identified for potential improvements?"
>
> We have some hypotheses, which are not completely verified.
> 1.  The STEM data mostly comes from textbooks and websites with a severe amount of markup language (HTML, Markdown, Latex, etc) and structures. Therefore, it requires a significant amount of effort to preprocess and normalize this type of corpus. The open LLMs haven't spent enough effort in dealing with this type of corpus.
> 2. The math-related problem-solving can benefit significantly from fine-grained human feedback. For example, OpenAI has released its human-feedback dataset at https://cdn.openai.com/improving-mathematical-reasoning-with-process-supervision/Lets_Verify_Step_by_Step.pdf. The open LLMs have yet to be benefited from such RL training.

---

### Official Review · Reviewer_quPx · 2023-08-07

**Soundness:** 3

**Excitement:**

4: Strong: This paper deepens the understanding of some phenomenon or lowers the barriers to an existing research direction.

**Paper Topic And Main Contributions:**

This work is aimed at evaluating the capabilities of LLMs to solve challenging math problems which may require domain-specific knowledge. Toward this objective, this work introduces TheoremQA. TheoremQA is a Question-Answering dataset where answering the question requires the application of appropriate theorems. The dataset consists of 800 high-quality questions, relying on 350 theorems across Maths, Physics, EE and CS, and Finance. Extensive experiments include comparisons among 16 LLMs with different sets of prompting strategies like Chain-of-Thought and Program-of-Thoughts. While GPT-4 was found to be significantly better than all the existing models, open-source models' performance was very poor.

**Questions For The Authors:**

It is mentioned that some hard-to-calculate questions and answers are omitted to build the dataset. What was the hypothesis of omitting those questions? Was it assumed that it would be more difficult for LLMs to solve those difficult questions?

Is there any implicit bias present in LLMs due to the corresponding training dataset and strategy used that affects the overall performance? Is there a way to determine that?

**Reasons To Accept:**

(1) The paper contributes a new dataset for the QA research community, which consists of university-level questions backed with corresponding Theorems
(2) It performs rigorous experiments with current SoTA LLMs with this dataset to check the performance measures of those LLMs
(3) Also, it gives insights into why LLMs fail in some particular cases to perform QA and shows a detailed error analysis and its limitations.

**Reasons To Reject:**

Although this paper proposed quality dataset questions along with the theorems, the distribution of the questions is somewhat imbalanced. Out of 800 questions, more than half of the questions are from Mathematics only. So, distribution is not even.

**Reproducibility:**

3: Could reproduce the results with some difficulty. The settings of parameters are underspecified or subjectively determined; the training/evaluation data are not widely available.

**Reviewer Confidence:**

3: Pretty sure, but there's a chance I missed something. Although I have a good feel for this area in general, I did not carefully check the paper's details, e.g., the math, experimental design, or novelty.

**Typos Grammar Style And Presentation Improvements:**

Suggest putting extra effort into improving the technical presentation. Needs significant effort.

---

> ### Author Rebuttal · Authors · 2023-08-25
>
> Thanks a lot for providing the constructive review.
>
> - "Although this paper proposed quality dataset questions along with the theorems, the distribution of the questions is somewhat imbalanced. Out of 800 questions, more than half of the questions are from Mathematics only. So, distribution is not even."
>
> This is an insightful observation. Currently, the dataset is driven by theorems, especially those requiring calculation, etc. It turns out that mathematics, including all its subareas, is the easiest area to find such theorems. Moving forward, we plan to expand our dataset continuously, either from our author's or the community's contributions. We will keep that in mind to balance the distribution across different STEM subjects.
>
> -  It is mentioned that some hard-to-calculate questions and answers are omitted to build the dataset. What was the hypothesis of omitting those questions? Was it assumed that it would be more difficult for LLMs to solve those difficult questions?
>
> In Line 2, we wrote, "During the annotation process, a small number of theorems were discarded due to their evaluation complexity". It's actually the hard-to-evaluate questions, rather than hard-to-calculate questions. For example, some theorem outputs are matrices, curves, functions, figures, etc. We don't have good tools to automatically evaluate the correctness of these outputs. That's why we have relatively fewer problems in functional analysis, complex analysis, graph theorems, etc.
>
> - What was the hypothesis of omitting those questions?
>
> The hypothesis is that the easy-to-evaluate questions are diversified enough to reflect LLMs' general capabilities to solve theorem-driven STEM questions.
>
> - Was it assumed that it would be more difficult for LLMs to solve those difficult questions?
>
> The hard-to-evaluate questions are not necessarily more difficult. Some of them are not very difficult, like SVD (Singular value decomposition), which outputs matrices as output. Some of them could be very difficult, like the ones from functional analysis.
>
> - Is there any implicit bias present in LLMs due to the corresponding training dataset and strategy used that affects the overall performance? Is there a way to determine that?
>
> Yes, due to the fact that almost all LLMs are based on text-only pre-training and instruction-tuning corpus (the multimodal parts are omitted during preprocessing). The model is not able to capture lots of non-textual signals in STEM theorems, like figures, graphs, tables, charts, or other imagery. The lack of multimodality in the pre-training corpus presents an implicit bias in the model to handle certain types of theorems better than others. For example, the numerical analysis theorems mostly reside in text format, so LLMs are generally good at them. In contrast, graph theorems contain lots of figures, which the models cannot handle very well.

---

### Meta-Review · Area_Chair_Yuuh · 2023-09-15

**Recommendation:** 4

**Metareview:**

The paper focusses on the description and evaluation of TheoremQA, a novel dataset designed to assess the capabilities of Large Language Models (LLMs) in applying theorems to solve complex scientific problems. TheoremQA is unique in its approach, as it requires the application of appropriate theorems to answer questions, making it a challenging benchmark for AI models. The dataset comprises 800 high-quality questions that are based on over 350 theorems spanning diverse fields such as Mathematics, Physics, Electrical Engineering & Computer Science, and Finance. The papers highlight extensive experiments conducted on 16 different LLMs using various prompting strategies. Among the models evaluated, GPT-4 outperformed others, while open-source models lagged in performance. The research provides valuable insights into the strengths and limitations of current AI models in scientific reasoning. Overall, TheoremQA serves as a significant benchmark that promises to drive further research in the domain.

The paper's introduction of TheoremQA, comprehensive evaluations, clear presentation, and valuable insights make it a significant contribution to the AI research community. In particular: a)  The paper introduces TheoremQA, a novel dataset designed for theorem-driven question-answering. It's the first dataset of its kind, covering a diverse range of theorems from fields like Mathematics, Physics, Electrical Engineering & Computer Science, and Finance. This dataset is poised to be a valuable resource for the research community. b) The authors have conducted an extensive evaluation of 16 state-of-the-art Large Language Models (LLMs) using TheoremQA. Their analysis is thorough, revealing both the strengths and limitations of current models in applying theorems and understanding multimodal inputs. c) The research employs innovative prompting strategies, such as Chain-of-Thoughts (CoT) and Program-of-Thoughts (PoT), to guide the LLMs in their evaluations.

However, one of the primary criticisms is the imbalanced distribution of questions within the dataset. Out of the 800 questions presented, a significant majority are from the Mathematics domain. This uneven distribution could limit the dataset's applicability and versatility across diverse fields, potentially skewing results or insights derived from it.

---

### Decision · Program_Chairs · 2023-10-07

**Decision:**

Accept-Main

**Comment:**

The paper focusses on the description and evaluation of TheoremQA, a novel dataset designed to assess the capabilities of Large Language Models (LLMs) in applying theorems to solve complex scientific problems. TheoremQA is unique in its approach, as it requires the application of appropriate theorems to answer questions, making it a challenging benchmark for AI models. The dataset comprises 800 high-quality questions that are based on over 350 theorems spanning diverse fields such as Mathematics, Physics, Electrical Engineering & Computer Science, and Finance. The papers highlight extensive experiments conducted on 16 different LLMs using various prompting strategies. Among the models evaluated, GPT-4 outperformed others, while open-source models lagged in performance. The research provides valuable insights into the strengths and limitations of current AI models in scientific reasoning. Overall, TheoremQA serves as a significant benchmark that promises to drive further research in the domain.

The paper's introduction of TheoremQA, comprehensive evaluations, clear presentation, and valuable insights make it a significant contribution to the AI research community. In particular: a)  The paper introduces TheoremQA, a novel dataset designed for theorem-driven question-answering. It's the first dataset of its kind, covering a diverse range of theorems from fields like Mathematics, Physics, Electrical Engineering & Computer Science, and Finance. This dataset is poised to be a valuable resource for the research community. b) The authors have conducted an extensive evaluation of 16 state-of-the-art Large Language Models (LLMs) using TheoremQA. Their analysis is thorough, revealing both the strengths and limitations of current models in applying theorems and understanding multimodal inputs. c) The research employs innovative prompting strategies, such as Chain-of-Thoughts (CoT) and Program-of-Thoughts (PoT), to guide the LLMs in their evaluations.

However, one of the primary criticisms is the imbalanced distribution of questions within the dataset. Out of the 800 questions presented, a significant majority are from the Mathematics domain. This uneven distribution could limit the dataset's applicability and versatility across diverse fields, potentially skewing results or insights derived from it.